# Lower airway microbiota compositions and diversity among ventilator-associated pneumonia patients across COVID-19 epidemic phases: a retrospective study

Shengyu Hao,[1] Chujun Zhou,[2] Yilin Wei,[1,3] Yuxian Wang,[1] Pan Jiang,[4] Jieqiong Song,[1] Ming Zhong[1]

**ABSTRACT**   Ventilator-associated pneumonia (VAP) is a major cause of morbidity in critically ill patients, and the SARS-CoV-2 infection has influenced the lung microbiome. This study aimed to examine the lower respiratory tract microbiome in VAP patients during different phases of the Shanghai COVID-19 epidemic. A total of 175 patients were included and divided into pre-epidemic (Pre), during-epidemic (Dur), and post-epidemic (Post) groups for analysis. Bronchoalveolar lavage fluid and serum were analyzed using next-generation sequencing. The intensive care unit (ICU) mortality rates were 48.3% (Pre group), 60.3% (Dur group), and 28.8% (Post group). Cytokine levels were lower in the Post group compared to the Pre group. *Acinetobacter*, *Candida*, and *Herpes Simplex Virus 1* (*HSV-1*) were the most frequently detected organisms. The prevalence of *Klebsiella pneumoniae*, *Enterococcus faecium*, *Aspergillus fumigatus*, and *HSV-1* was higher in the Dur group. α-Diversity of bacteria was significantly lower in the Dur group ($P < 0.05$), indicating reduced microbiome diversity. Multivariable Cox regression analysis identified APACHE II score (hazard ratio [HR] = 1.04, $P = 0.029$) and maximum bacterial load (HR = 1.67, $P = 0.046$) as independent risk factors for ICU mortality. This study highlights changes in microbiome composition across epidemic phases, which may inform treatment strategies.

**IMPORTANCE**   With the development of next-generation sequencing technology, it is increasingly being applied in clinical practice, especially in evaluating the prognosis of severe infections and co-infections. This study characterized the composition of microorganisms in the lower respiratory tract of ventilator-associated pneumonia (VAP) patients across three phases of the COVID-19 epidemic. Our study emphasizes that the relative abundance of bacteria, fungi, and viruses in bronchoalveolar lavage fluid of VAP patients may vary with the progression of the local epidemic and host immune status. These findings provide a mechanistic basis for optimizing targeted therapies in VAP management during infectious disease outbreaks.

**KEYWORDS**   ventilator-associated pneumonia, microbiome, species richness, next-generation sequencing technology, COVID-19

**Peer Reviewer** Daniel Fernández-Soto, Centro Nacional de Biotecnologia, Madrid, Spain

Address correspondence to Ming Zhong, zhongming2022@163.com, Jieqiong Song, song.jieqiong@zs-hospital.sh.cn, or Pan Jiang, jiang.pan@zs-hospital.sh.cn.

Shengyu Hao, Chujun Zhou, and Yilin Wei contributed equally to this article. The author order was determined by decreasing seniority.

The authors declare no conflict of interest.

See the funding table on p. 16.

Compared to other hospital departments, intensive care units (ICUs) have the highest rates of healthcare-associated infections. Notably, respiratory tract infections (RTIs) have been consistently identified as the most prevalent infections among ICU patients, accounting for 53.73%–64.7% (1, 2). Among the RTIs occurring in the ICU, ventilator-associated pneumonia (VAP) is a major clinical manifestation and one of the leading causes of ICU mortality (3, 4). Viral infections often impair both local and systemic immunity, leading to additional infectious complications. Furthermore, recent studies

have highlighted the critical role of the lower respiratory tract microbiota in immune system maturation and maintenance, which may affect patient outcomes (5).

COVID-19, caused by SARS-CoV-2, can progress to pneumonia and respiratory failure (6, 7). Coinfections are commonly observed in viral respiratory infections (8–10). Increasing evidence suggests that SARS-CoV-2 infection may alter the nasal, oropharyngeal, gut, and lung microenvironment, leading to microbial dysbiosis (11–13). This dysbiosis may predispose individuals to secondary infections or complications, contributing to higher morbidity and mortality rates among COVID-19 patients (14–17). Moreover, microbial dysbiosis may persist even after recovery from COVID-19 (18).

The global impact of COVID-19 continues to unfold, underscoring the need to investigate the relationship between respiratory microbiota and the pathogenesis of COVID-19 to develop improved diagnostic and management strategies. While most studies have focused on the relationship between microbiota and COVID-19 during the pandemic, as well as the long-term effects on the general population (19–23), there is limited research on changes in lung microbiota among critically ill patients with COVID-19. Recently, there have been growing advocacy and research efforts focused on the microbiome in the context of the COVID-19 pandemic (24). Due to the observations during and after the COVID-19 pandemic, there is a continued need for systematic research focusing on the characteristics and changes in the VAP among ICU patients. Additionally, examining the dynamics and compositions of co-existing pathogens is crucial for a better understanding of the microbial landscape in SARS-CoV-2-infected and post-infected patients, which can inform treatment strategies against those co-existing pathogens. This study employed metagenomic analysis of the changes in lung bacterial composition among ICU patients with VAP across three defined phases of Shanghai's COVID-19 epidemic, offering novel evidence to inform targeted antimicrobial and infection control strategies.

## MATERIALS AND METHODS

### Study population and study design

A retrospective analysis was conducted on patients aged 18 years and older who were admitted to the ICU at Zhongshan Hospital of Fudan University from 1 January 2021 to 1 October 2023 and diagnosed with VAP based on established criteria. Briefly, the clinical diagnostic criteria are as follows: (i) fever >38°C with no other cause, and (ii) leukocytosis or leukopenia and at least one of the following: (i) new onset or change in sputum; or (ii) cough, dyspnea, or tachypnea; or (iii) worsening gas exchange. The diagnostic criteria for radiology are as follows: chest radiographs or computed tomograms with evidence of pulmonary infiltrates or air bronchograms. If there is underlying pulmonary disease, it is necessary to compare with previous imaging to confirm new or progressive changes. The diagnostic criteria for microbiology are as follows: positive quantitative culture from a minimally contaminated lower respiratory tract specimen or positive sputum culture or non-quantitative lower respiratory tract culture (25). In our study, for patients who underwent endotracheal intubation in the ICU for more than 48 hours, if both clinical and radiology diagnostic criteria were met, we highly suspected VAP. Sputum culture and bronchoalveolar lavage fluid (BALF) were collected for testing at the same time. If either sputum culture or BALF was positive, the diagnosis of VAP was confirmed.

In this single-center study, all patients were endotracheally intubated, mechanically ventilated, and underwent fiberoptic bronchoscopy to collect BALF for both therapeutic purposes and next-generation sequencing (NGS) testing. The exclusion criteria were as follows: (i) age <18 years, (ii) pregnancy, and (iii) concurrent infections at other sites. Patients were classified into three groups based on the timing of Shanghai's first COVID-19 lockdown, their medical history, and clinical examination results: pre-epidemic group (Pre group): patients admitted to the ICU from 1 January 2021 to 28 February 2022 before the first outbreak in Shanghai with negative real-time PCR results and without any history of SARS-CoV-2 infection; during-epidemic group (Dur group): patients

admitted after the outbreak in Shanghai from 1 March 2022 to 31 December 2022, with a confirmed COVID-19 diagnosis based on positive PCR testing of nasopharyngeal and/or oropharyngeal samples; post-epidemic group (Post group): Patients admitted after the outbreak in Shanghai from 1 January 2023 to 1 October 2023, with a history of SARS-CoV-2 infection and negative PCR results upon current ICU admission. Clinical and laboratory data were collected from electronic medical records.

## Blood collection and sample processing

Blood samples were collected from the patients on the day of NGS testing. Approximately 2 mL of venous blood was drawn into EDTA anticoagulant tubes and transported at room temperature to clinical laboratories. D-dimer levels were measured using standardized automated procedures in the Department of Clinical Chemistry. C-reactive protein (CRP), procalcitonin (PCT), and ferritin concentrations were determined via immunoturbidimetric assays. Serum cytokines, including tumor necrosis factor-α (TNF-α), interleukin-1β (IL-1β), interleukin-2 receptor (IL-2R), interleukin-6 (IL-6), interleukin-8 (IL-8), and interleukin-10 (IL-10), were quantified by enzyme-linked immunosorbent assay (ELISA) using commercially available kits (Human ELISA Kit, R&D Systems, Minneapolis, MN, USA) according to the manufacturer's instructions.

Whole blood samples were processed via flow cytometry within 48 hours of collection. Prior to staining, leukocyte differential counts were performed to ensure lymphocyte concentrations remained within the linear detection range of the instrument. Lymphocyte subpopulations, including T cells (CD3$^+$), B cells (CD3$^-$CD19$^+$), helper/inducer T cells (Th, CD3$^+$CD4$^+$), cytotoxic T cells (Tc, CD3$^+$CD8$^+$), and natural killer cells (CD3$^-$CD16$^+$CD56$^+$), were analyzed using a FACSCalibur flow cytometer (BD Biosciences, San Jose, CA, USA). The BD Multitest 6-Color TBNK (Catalog No. 662967) included antibody cocktails as follows: BD Multitest CD3 FITC/CD8 APC-Cy7/CD45 PerCP-Cy5.5/CD4 PE-Cy7/CD16$^+$CD56$^+$ PE/CD19 APC. For whole-blood samples from patients collected in EDTA blood collection tubes, the procedure was carried out according to the manufacturer's instructions. Briefly, 20 µL of the diluted BD Multitest 6-Color TBNK reagent was pipetted to the bottom of the tube. Then, 50 µL of well-mixed anticoagulated whole blood was pipetted to the tube bottom, followed by incubation at room temperature in the dark for 30 min. Next, 450 µL of 1 × BD FACS Lysing Solution (Catalog No. 349202) was added to the tube. After shaking well, the tube was incubated in the dark at room temperature for 30 min. The samples were then analyzed by the flow cytometer. Unstained samples from the same blood specimen of the same patient were used as negative controls. The gating strategy and representative flow cytometry plots are provided in Fig. S1. The statistical analysis presents the percentage of each cell subset within the CD45$^+$ lymphocyte population, and the results are shown in Table S1.

## Sample collection and processing for NGS analysis

BALF was collected from patients according to standard procedure (26). Blood samples were collected via the veins with strict aseptic techniques. They were stored in EDTA tubes, left to stand at room temperature for 5 min, and then centrifuged at 1,600 × $g$ for 10 min at 4°C to separate the plasma. In critically ill ICU patients, pre-sampling antibiotic exposure represents a substantial confounding variable for NGS analysis. Prolonged diagnostic-therapeutic delays, defined as the interval between initial clinical suspicion of infection and targeted antibiotic administration, often necessitate empiric antibiotic regimens, thereby complicating microbiota interpretation. Notably, prior studies comparing NGS and conventional culture in antibiotic-treated samples have demonstrated that NGS retains pathogen detection capability even 24 hours after broad-spectrum antibiotic initiation (27, 28). To mitigate antibiotic effects on microbial profiling, our protocol adheres to the recommended practice of specimen collection within 24 hours of antibiotic administration. The standard operating procedure of the metagenomic workflow of BALF and blood samples could be found in previous studies (29, 30). Briefly, 1 mL of the sample was centrifuged at 12,000 × $g$ for 10 min to collect

the pathogens and human cells. Next, 50 µL of precipitate underwent depletion of host nucleic acid using 1 U of benzonase (Sigma) and 0.5% Tween 20 (Sigma) and incubated at 37°C for 5 min. A terminal buffer (400 µL) was added to stop the reaction. Then, the quantified unique DNA fragments (named unique molecular single-read identifiers) were spiked for each sample as an identity and internal control, which were PCR products of Oryza sativa 400–600 bp in length. A total of 600 µL of the mixture was transferred to new tubes containing 500 µL of ceramic beads for bead beating using a Minilys personal TGrinder H24 homogenizer (catalog number OSE-TH-01; Tiangen, China). Then, the nucleic acid from 400 µL of pretreated samples was extracted and eluted in 60 µL of elution buffer using a QIAamp UCP pathogen mini kit (catalog number 50214; Qiagen, Germany). The extracted DNA was quantified using a Qubit double-stranded DNA (dsDNA) high-sensitivity assay kit (catalog number Q32854; Invitrogen, USA).

## Library preparation and sequencing

Thirty microliter of the eluate was used to generate libraries using the Nextera DNA Flex kit (Illumina, San Diego, CA, USA) according to the manufacturer's instructions. Fragmentation and tagmentation of the DNA were performed using the bead-linked transposon. Dual indexing was conducted by employing the Integrated DNA Technologies for Illumina DNA/RNA UD indexes (catalog number 20027213). Purification and size selection were carried out following the double-sided bead purification procedure. A Qubit dsDNA HS assay kit was used to measure the library concentration. Library quality was assessed with an Agilent 2100 Bioanalyzer (Agilent Technologies, Santa Clara, CA, USA) using a high-sensitivity DNA kit. The library was prepared by pooling a 1.5 pM concentration of each purified sample equally for sequencing on an Illumina NextSeq 550 sequencer using a 75-cycle single-end sequencing strategy.

## Bioinformatic analysis

Microbial loads in this research were determined based on bioinformatic analysis of NGS data. For each batch of NGS workflows, blank samples are included and subjected to the identical procedures as the sequencing samples (from nucleic acid extraction to library preparation and sequencing). During data analysis, the NGS detection background is compared with this negative control to minimize the impact of environmental contamination. For analysis, Trimmomatic was used to remove low-quality reads, adapter contamination, duplicate reads, and reads shorter than 70 bp (31). Low-complexity reads were removed by Kcomplexity using default parameters. The human sequence data were identified and excluded by mapping to a human reference genome (hg38) using SNAP v1.0 beta.18. To construct the microbial genome database, pathogens and their genomes or assemblies were selected following the Kraken2 criteria for selecting representative assemblies for microorganisms (bacteria, viruses, fungi, protozoa, and other multicellular eukaryotic pathogens) from the NCBI Assembly and Genome databases (https://benlangmead.github.io/aws-indexes/k2). Microbial reads were aligned to the database using the Burrows-Wheeler Aligner software (32). We defined that reads with 90% identity of the reference were mapped reads. In addition, reads with multiple locus alignments within the same genus were excluded from the secondary analysis. Only reads mapped to the genome within the same species were considered.

We normalized the sequencing reads per ten million (RPTM) to eliminate the errors caused by various sequencing depths among samples. To establish the optimal threshold value for the >10 microbes with culture isolates, samples spiked with microbes were defined as positive samples, while the negative control was defined as the negative sample. Receiver operating characteristic curves were plotted for each target species using these samples. The parameter resulting in the highest area under the ROC curve was considered the positive cutoff value for this species. For microorganisms without culture isolates, the RPTM mean value and SD of this microorganism were calculated, and the RPTM (mean + 2 SDs) was set as a positive cutoff value (33).

To address potential batch effects arising from the multi-year study duration, several quality control measures were implemented. All samples were processed and sequenced at a single certified laboratory (Center for Infectious Diseases, Vision Medicals Co., Ltd, Guangzhou, China) using standardized protocols. Negative controls were included in each sequencing run to identify background microbial signatures from reagents and the laboratory environment; these background profiles were subsequently bioinformatically subtracted from the data of all patient samples. Furthermore, as previously detailed, a spiked internal standard was added to each sample to ensure technical consistency across different processing batches.

## Statistical analysis

Statistical analysis was performed using the R project (Version 4.3.1 for macOS). A $P$ value < 0.05 was considered statistically significant, and tests were two-sided. For comparisons between two groups of continuous variables, either Student's $t$-test or the Mann-Whitney $U$ test was used, depending on the data distribution. For comparisons among three groups, analysis of variance or the Kruskal-Wallis test was applied as appropriate. $\chi^2$ tests or Fisher's exact tests were used to compare categorical variables. Microbial α-diversity was assessed using the Shannon, Simpson, and evenness indices. These calculations were performed using the R project (Version 4.3.1 for macOS). To identify differentially abundant microbial features that could serve as potential biomarkers between groups, linear discriminant analysis effect size (LEfSe) was performed. A logarithmic linear discriminant analysis (LDA) score of 2.0 was considered the threshold for identifying statistically significant discriminative features. To assess the potential confounding effect of different antibiotic regimens on the microbiome analysis, these tests were also used to statistically compare the classes of antibiotics and combination therapies administered across the three groups. The Bonferroni method was applied for *post hoc* comparisons. Missing data were handled using multiple imputation, and Cox proportional hazards regression analysis was performed to evaluate the effect of multiple factors on the risk of outcome events. The results of the Cox regression analysis are presented as forest plots. We used the maximum selection ranking statistics (maxstat) method to determine the optimal cutoff point for the maximum bacterial load and converted it into a binary variable before re-analyzing the Cox regression analysis.

## RESULTS

### Patient demographics and group characteristics

From 1 January 2021 to 1 October 2023, 10,515 patients were admitted to the ICU, 3,940 patients underwent mechanical ventilation for more than 48 hours, and 258 of these patients were diagnosed with VAP. According to the exclusion criteria, 83 patients were excluded: 11 were <18 years old, 8 were pregnant, and 64 had concurrent infections at other sites. Therefore, 175 patients were included in this study. Patients were stratified into three groups based on admission timing and SARS-CoV-2 infection status. The Pre group ($n$ = 58, 33.1%) included individuals admitted to the ICU prior to Shanghai's first confirmed SARS-CoV-2 outbreak, with negative SARS-CoV-2 PCR results and no documented history of infection. The Dur group ($n$ = 58, 33.1%) consisted of patients admitted during the local outbreak period, who tested positive for SARS-CoV-2 via nasopharyngeal/oropharyngeal PCR. The Post group ($n$ = 59, 33.8%) comprised patients admitted following the outbreak's resolution, with a history of SARS-CoV-2 infection but negative PCR results upon current ICU admission. Among them, 80 (45.7%) patients succumbed in the ICU, while 95 (54.3%) survived and were discharged to the ward. The mortality rates in the three groups were 48.3%, 60.3%, and 28.8%, respectively. As shown in Table S1, the median age of patients was 66 years, with the Dur group being older than the Pre and Post groups ($P$ < 0.001), and deceased patients were older in both the Pre ($P$ < 0.01) and Dur groups ($P$ < 0.05). There were no differences in APACHE II ($P$ = 0.225) and SOFA ($P$ = 0.710) scores among the three groups; however, within each group, the SOFA scores of deceased patients were higher than those of survivors ($P$ < 0.001 in

the three groups). The Charlson score in the Dur group was higher than that in the Pre group ($P < 0.01$), but there was no difference between survivors and deceased patients within each group. The usage rates of steroids or immunosuppressive drugs were higher in the Post group compared to the Pre group ($P < 0.05$). Additionally, the rate of CRRT treatment in deceased patients was higher than that in survivors in the Post group ($P < 0.01$). No significant differences were observed among the three groups regarding gender, body mass index (BMI), length of hospital stay, or ICU admission duration.

To compare the changes in immune response in these VAP patients, pairwise comparisons were conducted among the three groups in cytokines, leukocytes and their subsets, platelets, and inflammatory factors (Table S1). No significant increase in cytokines in the Dur group was observed compared with the Pre group. Nevertheless, the levels of TNF-α, IL-8, and ferritin in the Pre and Dur groups were found to be elevated compared with the Post group. Meanwhile, the levels of IL-1β, IL-2R, and IL-10 were higher in the Pre group than in the Post group. PCT levels were significantly lower in the Dur group compared to the Pre and Post groups. No differences were observed in IL-6 levels, WBC and subsets, PLT, D-dimer, and CRP among the three groups. In the Pre and Dur groups, TNF-α, IL-1β, IL-2R, IL-6, and IL-8 levels were higher in the deceased patients compared to the survivors, while PLT levels were significantly lower in the deceased patients. Notably, in the Post group, except for CD56 and CRP, which were elevated in the non-survivor patients, no significant differences were observed between the survivor and non-survivor groups (Table S1).

Furthermore, to control for antibiotic use as a confounding factor, the distribution of antibiotic classes among the groups was analyzed. Statistical analysis revealed no significant differences among the three groups in the use of most individual antibiotic classes. The most frequently used antibiotics were carbapenems (56.9%, 53.4%, and 55.93% in the Pre, Dur, and Post groups, respectively; $P = 0.781$), third-generation cephalosporins (20.69%, 27.6%, and 20.34%; $P = 0.643$), and glycopeptides (17.2%, 13.8%, and 30.51%; $P = 0.061$). Notably, while a significant difference was found in the overall comparison of combination therapy ($P = 0.048$), subsequent pairwise comparisons showed no significant differences between the groups. These results indicate that antibiotic exposure was largely comparable across the study groups (Table S2).

## Bacterial detection and diversity analysis in BALF samples across different phases

Heatmaps were generated based on the sequence read counts of microorganisms detected in 149 serum samples (Fig. 1A) and 143 BALF samples (Fig. 1B) from patients, analyzed by NGS. The diversity of detected bacteria, viruses, and fungi was greater in BALF compared to serum, with counts of 35 vs 25 for bacteria, 22 vs 19 for viruses, and 19 vs 7 for fungi, respectively. We further analyzed the microorganisms in BALF. Relative abundance detection revealed that the three most abundant genera in the Pre group were *Acinetobacter* (26.07%), *Pseudomonas* (10.93%), and *Stenotrophomonas* (8.12%). In the Dur group, the top three genera shifted to *Acinetobacter* (38.29%), *Klebsiella* (15.45%), and *Enterococcus* (10.27%). In the Post group, the most abundant genera were *Acinetobacter* (44.24%), *Streptococcus* (12.19%), and *Stenotrophomonas* (7.61%; Fig. 2A and B). The most common species in the Pre group were *Acinetobacter baumannii* (23.07%), *Pseudomonas aeruginosa* (11.09%), and *Stenotrophomonas maltophilia* (10.05%); *Acinetobacter baumannii* (31.47%), *Klebsiella pneumoniae* (12.26%), and *Stenotrophomonas maltophilia* (6.76%) were the predominant species detected in the Dur group. Notably, the abundance of *Klebsiella pneumoniae* in the Dur group (12.26%) was significantly higher than that in the Pre (3.1%) and Post groups (6.64%). Interestingly, *Enterococcus faecium* was rarely detected in the Pre group (0.89%) but exhibited higher abundance in the Dur (6.57%) and Post group (4.50%; Fig. 2C and D). This may be related to the intestinal barrier dysfunction caused by the *SARS-CoV-2* infection. To further quantify the differences in the microbiome, the alpha diversity and beta diversity of pulmonary microbes in the Pre, Dur, and Post groups were compared.

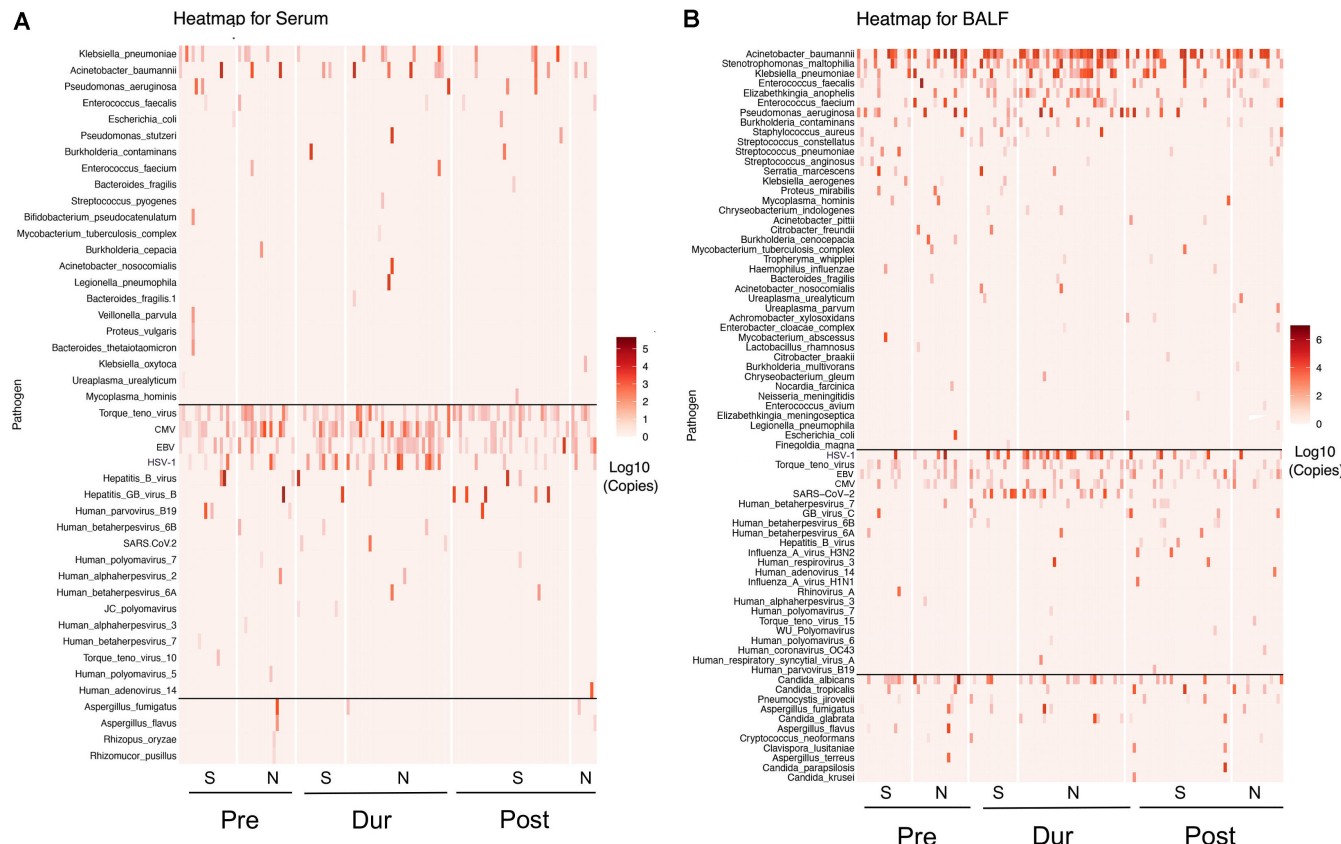

**FIG 1** Heatmaps showing the sequence read counts of all patients' pathogens detected by NGS in serum (A) and BALF (B), grouped according to COVID-19 prevalence and survival status. S, survivor; N, non-survivor.

Parameters, including Shannon, Simpson, and evenness indices, were calculated. The analysis revealed that the lung microbial diversity in the Pre group was higher than that in the Dur and Post groups, with a trend of recovery observed in the Post group (Fig. 2E and F).

## Fungal detection in BALF samples across different phases

Nineteen fungal species were identified in the BALF from these patients, with *Candida* being the most prevalent genus (40.00%). In the Pre group, the most frequently detected fungal genera were *Candida* (41.80%), *Malassezia* (20.12%), and *Corynespora* (8.36%). In the Dur group, the predominant genera were *Candida* (43.40%), *Corynespora* (10.38%), and *Aspergillus* (8.18%). In the Post group, the most common genera included *Candida* (34.69%), *Trichosporon* (20.94%), and *Corynespora* (10.63%; Fig. 3A and B). Regarding fungal species, the relative abundance of *Candida albicans* in the Dur group (33.63%) was higher than that in the Pre group (30.01%) and the Post group (18.70%). Notably, *Trichosporon akiyoshidainum* increased significantly in the Post group, rising from 2.81% in the Pre group to 21.21% in the Post group and 5.4% in the Dur group. *Aspergillus fumigatus* showed a similar trend, with its abundance being 1.11% in the Pre group, 5.1% in the Dur group, and 4.80% in the Post group. In contrast, *Malassezia restricta* decreased significantly from the Pre group (16.48%) to the Dur (2.00%) and Post groups (2.66%; Fig. 3C and D).

## Viral detection in BALF samples across different phases

In the BALF of ICU patients, a total of 23 viruses were detected. In the Pre group, the three most prevalent viruses were *Torque teno virus* (26.33%), *Cytomegalovirus* (*CMV*) (11.80%),

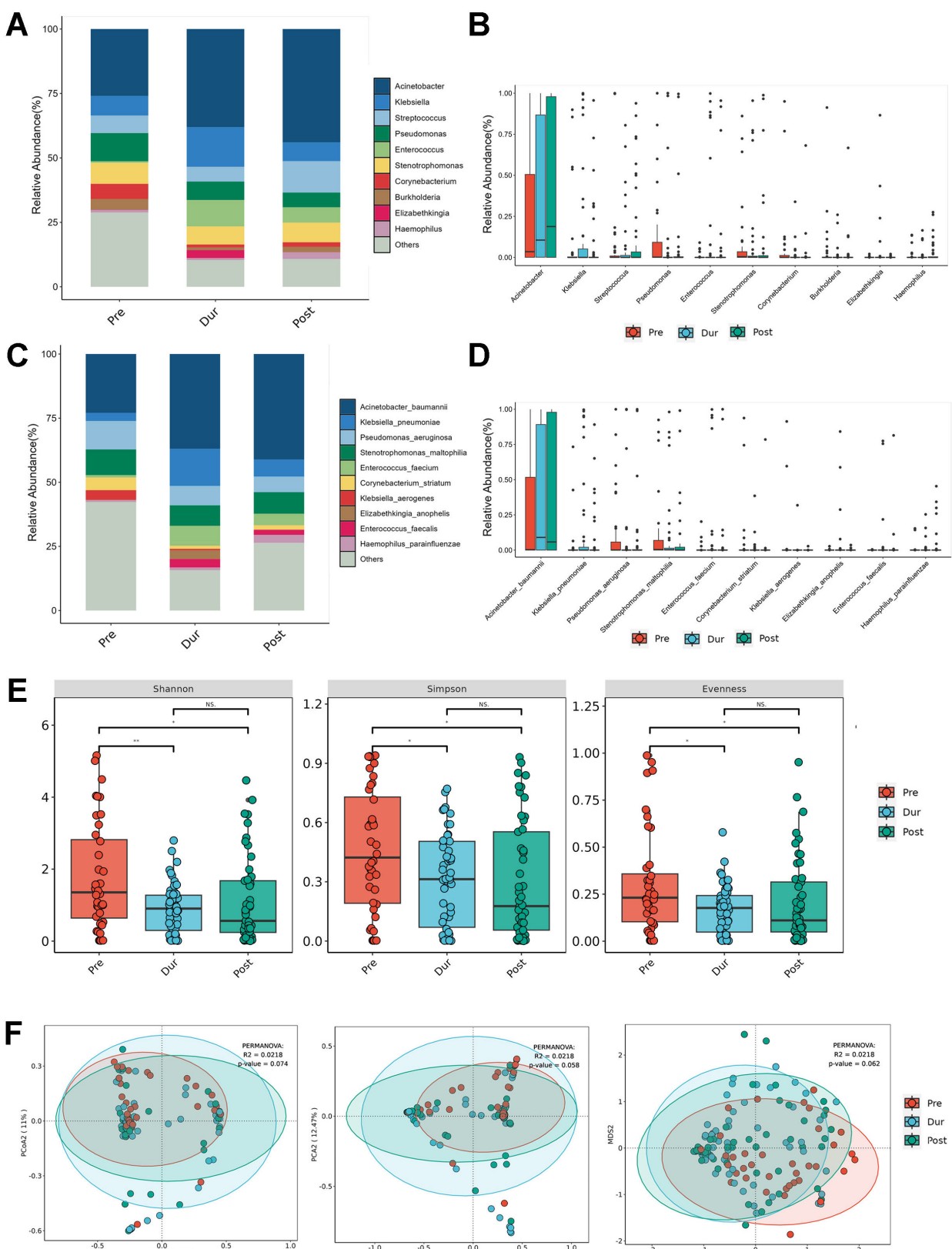

**FIG 2** Temporal shifts in lower airway bacteriological compositions across different phases of the COVID-19 epidemic. (A) Composition of microorganisms at the genus level. The bar chart was drawn based on the relative abundance of the corresponding microorganisms at the genus level. (B) Comparison of the top 10 microorganisms in the relative abundance at the genus level. The relative abundance of the top 10 microorganisms at the genus level was compared by the

Fig 2 (Continued)

Wilcoxon-Mann-Whitney rank-sum test, and a box plot was drawn. (C) Composition of microorganisms at the species level. The bar chart was drawn based on the relative abundance of the corresponding microorganisms at the species level. (D) Comparison of the top 10 microorganisms in relative abundance at the species level. The relative abundance of the top 10 microorganisms at the species level was compared by the Wilcoxon-Mann-Whitney rank-sum test, and a box plot was drawn. (E) α-Diversity by Shannon, Simpson, and evenness indices. (F) β-Diversity based on principal component analysis (PCA), Bray-Curtis principal coordinate analysis (PCoA), and Bray-Curtis non-metric multidimensional scaling (NMDS) analysis between the Pre, Dur, and Post groups. A *P* value was obtained by the Wilcoxon-Mann-Whitney rank-sum test. *P* values less than 0.05 were considered statistically significant.

and *Herpes Simplex Virus 1* (*HSV-1*; 11.06%). In the Dur group, the top three viruses were *HSV-1* (35.94%), *SARS-CoV-2* (30.33%), and *Epstein-Barr virus (EBV*; 11.29%). In the Post group, the most common viruses were *Torque teno virus* (17.65%), *HSV-1* (17.28%), and *EBV* (13.74%). We observed a significant increase in the relative abundance of *HSV-*1 in the Dur group, while the *Torque teno virus* showed a notable decrease. Additionally, we noted that the prevalence of *Influenza A virus H1N1* and *Influenza A virus H3N2* rose from 0% in the Pre group to 3.25% and 2.73%, respectively, in the Post group (Fig. 3E and F). LEfSE analysis determined the discriminative microbial features between the Pre, Dur, and Post groups. As shown in Fig. 4A, *Pseudomonas aeruginosa*, *HSV-1*, and *Streptococcus dysgalactiae* were identified as enriched species in the Pre, Dur, and Post groups, respectively.

## Microbiota analysis and mortality risk factors in surviving and deceased patients

We further analyzed the microbiota of surviving and deceased patients. According to the relative abundance, the most common genera in the survivor group were *Acinetobacter* (31.61%), *Streptococcus* (11.76%), and *Klebsiella* (10.32%). At the same time, the most common genera in the non-survivor group were *Acinetobacter* (42.47%), *Klebsiella* (10.55%), and *Pseudomonas* (9.11%; Fig. 5A and B). At the species level, the most prevalent bacteria in the survivor group were *Acinetobacter baumannii* (28.77%), *Stenotrophomonas maltophilia* (10.98%), and *Klebsiella pneumoniae* (7.82%). Different from the survivor group, the most frequent species in the non-survivor group were *Acinetobacter baumannii* (51.08%), *Klebsiella pneumoniae* (11.85%), and *Pseudomonas aeruginosa* (11.58%). Significant increases were observed in *Acinetobacter baumannii* (survivor group vs non-survivor group: 28.77% vs 51.08%) and *Enterococcus faecium* (survivor group vs non-survivor group: 1.87% vs 9.61%) in the non-survivor group (Fig. 5C and D). However, the alpha-diversity and beta-diversity showed no difference between the survivor and non-survivor groups (Fig. 5E and F). Discriminative microbial features between the survival and non-survival groups were determined by LEfSE. *Elizabethkingia anophelis* was a characteristic bacterium in the non-survivor group, while *Streptococcus pseudopneumoniae* was identified as an enriched species in the survivor group (Fig. 4B).

To evaluate the impact of multiple factors on the mortality risk, a Cox proportional hazards regression analysis was performed. Using stepwise selection, 15 variables (age, gender, APACHE II, BMI, COVID-19 status, the usage rates of steroids or immunosuppressive drugs, total number of infectious pathogens, maximum bacterial load, maximum viral load, maximum fungal load, and the presence or absence of *Acinetobacter baumannii*, *Enterococcus*, *HSV-1*, *EBV*, and *CMV*) were screened, and the final model retained three variables: age (HR = 1.01, 95% CI [0.99, 1.03], *P* > 0.05), APACHE II score (HR = 1.03, 95% CI [1.00, 1.07], *P* = 0.028), and maximum bacterial sequence count (HR = 1.00, 95% CI [1.00, 1.00], *P* < 0.001). Despite the highly significant *P*-value for the maximum bacterial sequence count, its hazard ratio is close to 1, indicating that each unit increase in bacterial sequence count has a minimal effect on survival time. Therefore, we used the Maxstat method to determine the optimal cutoff for maximum bacterial sequence count at 167314 (Fig. S2) and recategorized it as a binary variable for subsequent Cox regression analysis.

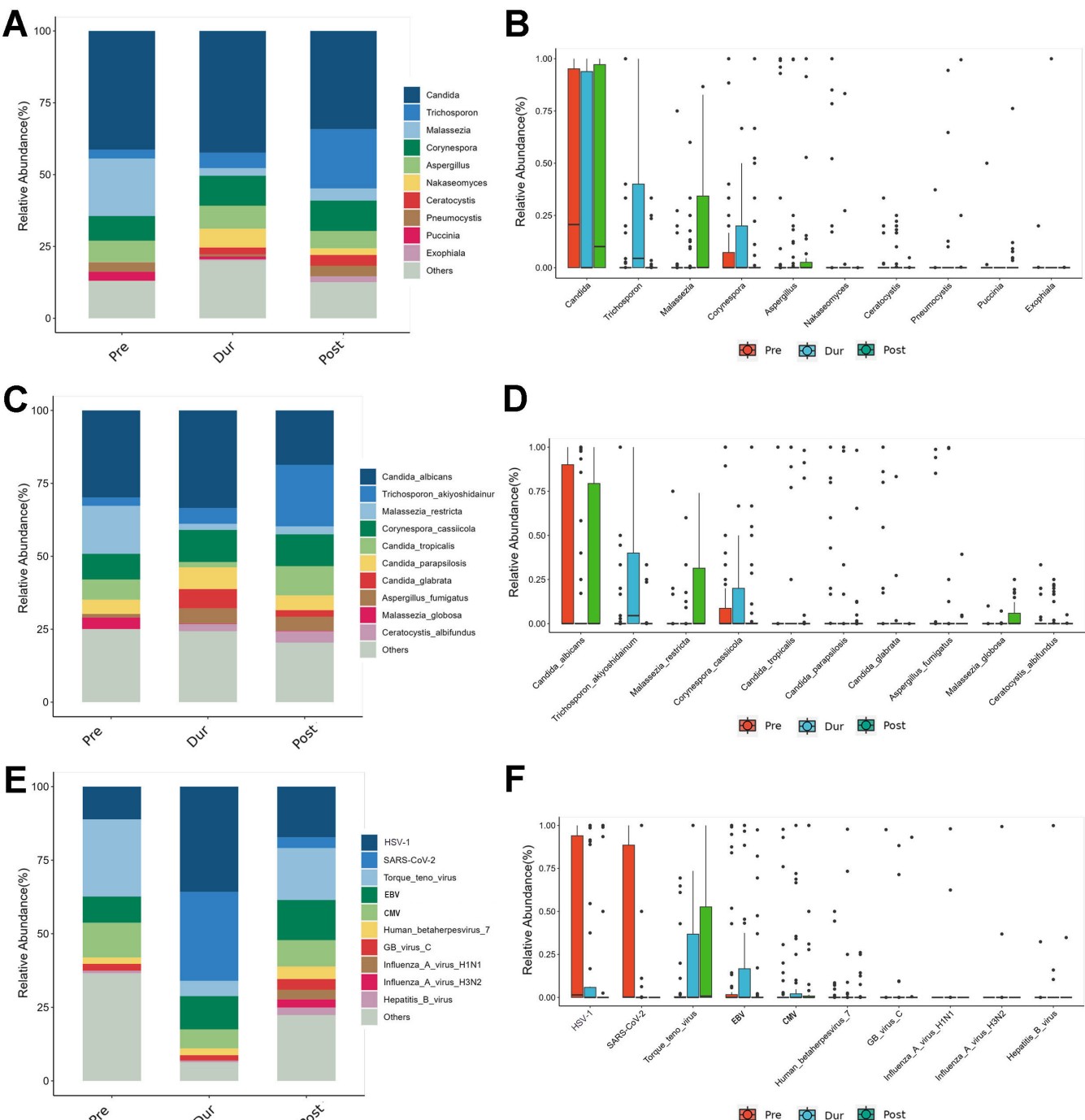

**FIG 3** Lower airway fungal and virus compositions of the Pre, Dur, and Post groups. (A) Composition of fungus at the genus level. The bar chart was drawn based on the relative abundance of the corresponding fungus at the genus level. (B) Comparison of the top 10 fungi in the relative abundance at the genus level. The relative abundance of the top 10 fungi at the genus level was compared by the Wilcoxon-Mann-Whitney rank-sum test, and a box plot was drawn. (C) Composition of fungus at the species level. The bar chart was drawn based on the relative abundance of the corresponding fungus at the species level. (D) Comparison of the top 10 fungi in relative abundance at the species level. The relative abundance of the top 10 fungi at the species level was compared by the Wilcoxon-Mann-Whitney rank-sum test, and a box plot was drawn. (E) Comparison of the top 10 viruses in relative abundance. The relative abundance of the top 10 viruses was compared by the Wilcoxon-Mann-Whitney rank-sum test, and a box plot was drawn. (F) Comparison of the top 10 viruses in relative abundance. The relative abundance of the top 10 viruses was compared by the Wilcoxon-Mann-Whitney rank-sum test, and a box plot was drawn.

The adjusted model retained four variables: age, gender, APACHE II score, and maximum bacterial sequence count greater than 167314. The results, illustrated in a

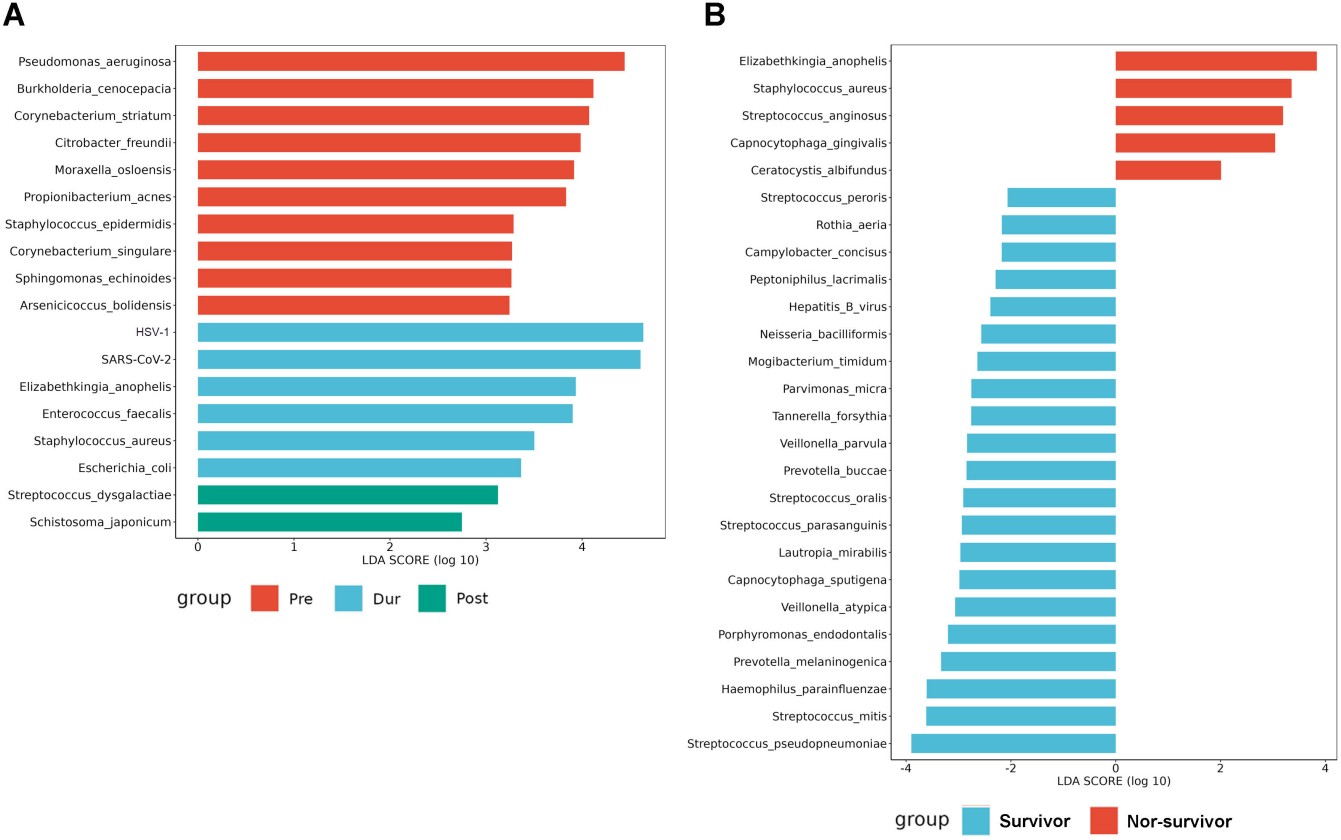

**FIG 4** The LEfSe. (A) The differentially abundant microbiome between the Pre, Dur, and Post groups. (B) The differentially abundant microbiome between the survivor and non-survivor groups. Bar plots show LDA scores of each genus. The LDA scores indicate the effect sizes of each genus, and genera with an LDA score ≥2 are shown.

forest plot, showed that both APACHE II score (HR = 1.04, 95% CI [1.00, 1.07], *P* = 0.029) and maximum bacterial sequence count (HR = 1.67, 95% CI [1.01, 2.77], *P* = 0.046) were significantly associated with the risk of the outcome event (Fig. 6). These findings suggest that higher APACHE II scores and a maximum bacterial sequence count greater than 167314 may significantly increase the risk of patient mortality. Although age (HR = 1.02, 95% CI [1.00, 1.03], *P* = 0.064) and gender (HR = 0.64, 95% CI [0.37, 1.12], *P* = 0.121) did not reach statistical significance, the effect of age approached significance. Future studies with larger sample sizes could further explore the potential impact of age on risk.

## DISCUSSION

The COVID-19 pandemic has entered a stable stage and is gradually being managed as an endemic disease similar to influenza viruses due to the effective control measures implemented by the government. Meanwhile, the long-term effects of COVID-19 infection have increasingly drawn the attention of researchers. Severe patients may experience pneumonia due to COVID-19 infection or be admitted to the ICU with a history of prior infection. However, currently, there are few studies on the changes in pathogenic microorganisms in ICU patients before and after the COVID-19 epidemic. Coinfections and superinfections are common in critically ill patients and are associated with poor prognosis (34). Due to the changes in the immune function, respiratory barrier, and lung microenvironment of COVID-19 patients, the spectrum of infectious pathogens may be different (34, 35). This study investigated the microbiota compositions of VAP patients in the ICU before, during, and after the COVID-19 epidemic by NGS analysis.

In critically ill patients, the emergence of superinfections caused by bacteria, fungi, or viruses substantially complicates the clinical course (35). The advent of NGS has

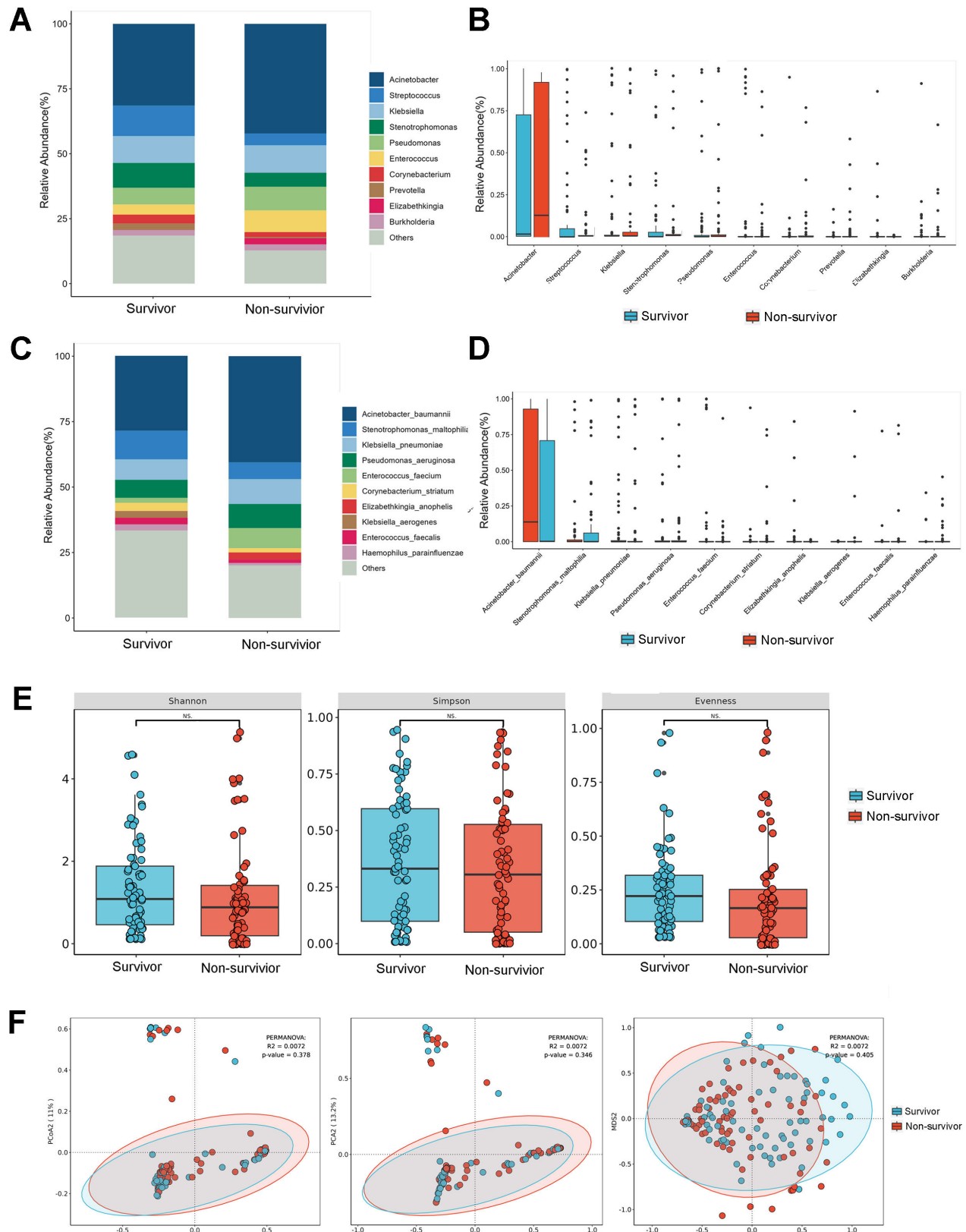

**FIG 5** Lower airway bacteriobiota compositions of survivor and non-survivor groups. (A) Composition of microorganisms at the genus level. The bar chart was drawn based on the relative abundance of the corresponding microorganisms at the genus level. (B) Comparison of the top 10 microorganisms in the relative abundance at the genus level. The relative abundance of the top 10 microorganisms at the genus level was compared by the Wilcoxon-Mann-Whitney

Fig 5 (Continued)

rank-sum test, and a box plot was drawn. (C) Composition of microorganisms at the species level. The bar chart was drawn based on the relative abundance of the corresponding microorganisms at the species level. (D) Comparison of the top 10 microorganisms in relative abundance at the species level. The relative abundance of the top 10 microorganisms at the species level was compared by the Wilcoxon-Mann-Whitney rank-sum test, and a box plot was drawn. (E) α-Diversity analyzed by Shannon, Simpson, and evenness indices. (F) β-Diversity based on PCA, Bray-Curtis PCoA, and Bray-Curtis NMDS analysis between the survivor and non-survivor groups. *P* value was obtained by the Wilcoxon-Mann-Whitney rank-sum test. *P* values less than 0.05 were considered statistically significant.

surmounted the drawbacks of prolonged cultivation time and low sensitivity associated with traditional methods, thereby permitting the concurrent detection of multiple microorganisms. Its accuracy and clinical application value are being continuously validated and corroborated, with studies demonstrating diagnostic performance for respiratory pathogens comparable to targeted quantitative polymerase chain reaction (qPCR) assays (36–39).

In the ICUs of the United States, the prevalent etiology factors associated with VAP were ascertained and ranked in descending order of frequency as follows: *Staphylococcus aureus*, succeeded by *Pseudomonas aeruginosa*, and then *Klebsiella pneumoniae*. Conversely, within the European ICUs, the pathogens most commonly responsible for VAP were manifested in the order of *Klebsiella pneumoniae*, trailed by *Staphylococcus aureus* and subsequently *Pseudomonas aeruginosa* (40). In our research, *Acinetobacter baumannii*, *Klebsiella pneumoniae*, and *Pseudomonas aeruginosa* were identified as the most common pathogens causing VAP, which is consistent with the results of a recent nationwide VAP survey in China (41). Although previous studies suggested that the etiology of VAP does not differ between COVID-19 and non-COVID-19 patients (42), our single-center analysis identified distinct pathogen profiles among COVID-19 patients admitted to a Shanghai hospital across epidemic waves. The relative abundance of *Klebsiella pneumoniae* and *Enterococcus faecium* showed significant changes across different groups, with *Enterococcus faecalis* notably increasing during the acute phase of the local COVID-19 outbreak and persisting at elevated levels post-epidemic peak. Additionally, microbial diversity exhibited a trend of initial decline followed by recovery in the Pre, Dur, and Post groups. These findings suggest that COVID-19 infection disrupted the original microbial balance, impaired pulmonary microbial diversity, and may have triggered gut microbiota displacement.

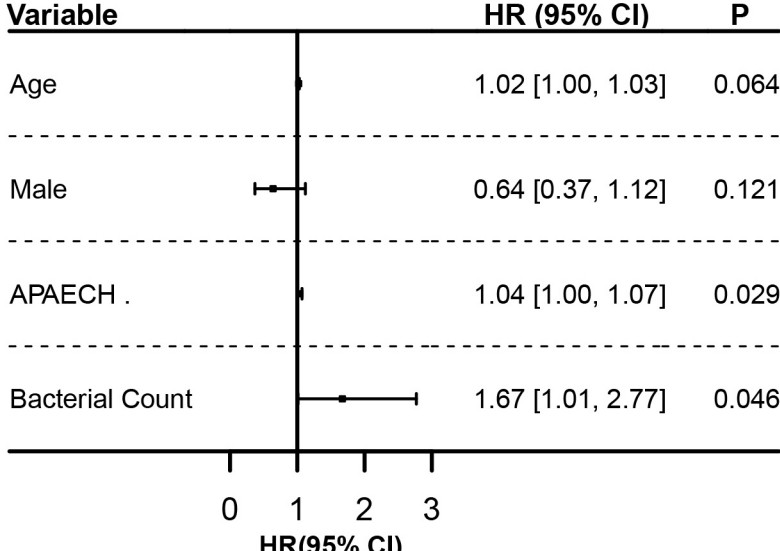

**FIG 6** Forest plot analysis for ICU mortality of the VAP patients.

Recent studies highlighted the interplay between the gut microbiota and the lungs, often referred to as the "gut-lung axis" (12, 43). SARS-CoV-2 can directly or indirectly induce gut microbiota dysbiosis. For instance, viral infection triggers the inflammatory response, prompting the release of cytokines, such as IL-6 and TNF-α (44). These mediators can disrupt the balance of the gut microbiota and impair the gut barrier. Furthermore, the resulting dysbiosis and microbial translocation can exacerbate the inflammatory response, creating a vicious cycle. In this study, however, we did not observe increased inflammation or evidence of immune tolerance in the Dur group, which appears to be inconsistent with previous studies. Specifically, there was no elevation in cytokine levels in the Dur group compared to the Pre group, and no statistically significant differences were found in leukocyte counts or their sub-population. In the intra-group analysis, we observed that inflammation levels in the non-survivor subgroups were significantly higher than those in the survivor subgroups in both the Pre and Dur groups. Interestingly, in the Post group, the inflammatory response seemed to be more subdued, with cytokine levels generally lower than those in the Pre and Dur groups. Moreover, except for the increased levels of CRP and CD56, no other significant changes in cytokine levels were observed between the survivor and non-survivor subgroups. We hypothesize that patients who have been affected by and recovered from COVID-19 may experience immune exhaustion or tolerance. This hypothesis has also been validated in some basic research studies. Cheong et al. demonstrated that months to a year after severe COVID-19 infection, persistent alterations in the innate immune phenotype and epigenetic programming of hematopoietic stem cells and progenitors occurred within the immune system. These changes led to impaired tissue repair and a diminished capacity to combat subsequent infections, reflecting immune dysfunction and a state akin to immune tolerance following severe infection, which in turn resulted in reduced resistance to other pathogens (45).

We further described the fungal detection in the BALF of these patients. Although NGS results cannot definitively indicate whether Candida species are invasive or colonizing, Candida, particularly Candida albicans, was the most frequently detected fungus in this research. Moreover, the relative abundance of Candida albicans was highest in the Dur group. This finding is consistent with previous studies, which have reported that the incidence of Candida bloodstream infections in COVID-19 patients is 2–10 times higher than in non-COVID-19 patients (46). Candida albicans is the most commonly reported yeast species in critically ill COVID-19 patients, accounting for 44% of Candida bloodstream infection cases in a multicenter study conducted in the United States (47). In early 2020, China reported the first case of COVID-19-related pulmonary aspergillosis (48). Since then, several case series and cohort studies have highlighted the importance of this potentially life-threatening secondary infection. In this study, we also observed a significant increase in the detection of Aspergillus species in the Dur and Post groups.

Recent studies have shown that COVID-19 infection can trigger the reactivation of herpesvirus species, leading to the persistent presence of the virus as a co-infection with COVID-19 (49, 50). To date, eight herpesvirus types have been identified. The most prevalent ones among them are HSV-1, HSV-2, Varicella-Zoster Virus, HSV-6, EBV, and CMV (51). Published research has demonstrated that viral reactivation or co-infection in COVID-19 patients can result in markedly elevated mortality rates among critically ill patients. An observational cohort study found that HSV reactivation is common in COVID-19 patients receiving mechanical ventilation, while CMV reactivation is less frequent. Furthermore, the dynamic viral load of HSV was associated with an increased 90-day mortality rate (52). In this study, HSV-1, EBV, and CMV were identified in BALF with elevated relative abundances. Notably, the relative abundance of HSV-1 was significantly higher in the Dur group compared to the Pre and Post groups. Interestingly, the relative abundance of CMV demonstrated a downward tendency in the Dur group. Although our study did not demonstrate a direct association between the detection of EBV, HSV,

and an increased risk of mortality, this may be limited by the retrospective design, small sample size, and the inability to confirm whether these viruses were reactivated.

Studies have shown that changes in immune status after COVID-19 infection will directly affect the host microbiome, especially the diversity of pathogens (53, 54). The acute stage of COVID-19 infection and the immune response it triggers may lead to the loss of the host's normal microbiome and even an imbalance of healthy flora through mechanisms such as the use of antibiotics, inflammatory response, and oxidative stress (55, 56). The analysis revealed that the lung microbial diversity in the Pre group was higher than that in the Dur and Post groups, with a trend of recovery observed in the Post group. In addition, immune dysregulation or immune escape after COVID-19 infection may provide favorable conditions for the proliferation of specific pathogens. Certain pathogens, such as *Streptococcus pneumoniae*, *Staphylococcus aureus*, and fungi (such as *Candida albicans*), may gain reproductive advantages when the immune system is unbalanced (57). COVID-19 infection may lead to viral-bacterial co-infection, further altering the diversity of bacterial communities (54).

As a high-throughput technology, NGS has been widely used in the diagnosis and monitoring of infectious diseases in recent years. Bacterial load refers to the number or concentration of bacteria present at the site of infection or in the body and is generally considered to be a key factor affecting clinical outcomes. For example, in critically ill patients, such as sepsis and severe pneumonia, higher bacterial loads are often associated with poor clinical outcomes (58–60). Our study found that a maximum bacterial sequence count greater than 167314 was a significant mortality risk factor. The pathogen load not only affects the clinical manifestations of the disease but is also closely related to the patient's immune response. Studies have shown that patients with immunodeficiency or immunosuppression usually have higher bacterial loads, and these patients are often unable to effectively clear pathogens from the body, leading to worsening of the disease. In addition, excessive bacterial loads may also lead to excessive immune responses, thereby exacerbating inflammatory responses and inducing secondary damage, further worsening the prognosis (61, 62).

While our findings demonstrate significant shifts in the lower airway microbiota across the pandemic phases, these results must be interpreted within the context of multiple, co-evolving clinical and epidemiological factors. Although ICU admission criteria could have theoretically shifted, the overall baseline severity of illness was comparable across the cohorts, as evidenced by non-significant differences in APACHE II and SOFA scores. However, critical demographic and clinical differences likely acted as confounders. Specifically, patients in the during-epidemic group were older and had a higher comorbidity burden as measured by the Charlson score, while the Post group had significantly higher rates of steroid or immunosuppressive drug use, all of which can independently shape the airway microbiome. Regarding antimicrobial pressure, our analysis revealed that the use of most individual antibiotic classes was largely comparable across the groups, suggesting that this may not have been the primary driver of the observed microbial changes. Furthermore, pandemic-related surges may have impacted nosocomial transmission dynamics for common VAP pathogens (63). Therefore, while SARS-CoV-2 infection appears to be a key factor, particularly in reducing microbial diversity, the observed shifts in the VAP microbiome must be understood as a multifactorial phenomenon reflecting a complex interplay between the virus, evolving patient characteristics, clinical practices, and the ICU environment.

Acknowledging the study's limitations is crucial. First, the presentation solely focuses on the findings of the examination into the presence of pathogenic microorganisms in the lower respiratory tract, leaving the judgment of whether they are pathogenic or colonized bacteria to clinicians. Moreover, orthogonal validation of these pathogens is also crucial. Future studies will integrate qPCR or other targeted methodologies to confirm the presence and load of pathogens, thereby aiming to provide more rigorous evidence for their clinical significance. Second, as a single-center retrospective study, it is subject to potential selection bias. Although Pre and Dur groups were comparable,

the Post cohort had a distinct ICU admission etiology profile compared to the other two groups, potentially confounding microbiota analysis. Prospective and multi-center studies are necessary for further confirmation. Third, NGS technology has some inherent flaws, with detection results prone to contamination by the kit-related background microorganisms and nucleic acids. Advancements and updates in NGS technology are expected to address this issue.

## Conclusion

In conclusion, our study demonstrates that the relative abundance of bacteria, fungi, and viruses in BALF may change as the epidemic progresses. However, the impact of SARS-CoV-2 infection and its aftermath on critically ill patients still requires further investigation. In the future, as sequencing technology advances and costs decrease, NGS could become a routine diagnostic and monitoring tool, particularly in assessing the prognosis of severe infections and co-infections. Prospective studies with standardized and comprehensive microbiological sampling before the initiation of antimicrobial treatment are needed to better characterize co-infections and secondary infections in COVID-19 patients.

## ACKNOWLEDGMENTS

We would like to thank Vision Medicals for providing the NGS test.

S.H., J.S., P.J., and M.Z. conceived and designed the study. S.H., C.Z., and Y. Wei drafted the manuscript and analyzed the data. J.S., Y. Wang, and C.Z. were involved in the data collection and drafted the manuscript. All authors commented on previous versions of the manuscript. All authors read and approved the final manuscript.

This research was sponsored by the National Science Fund for Young Scholars (82200061), the Shanghai Sailing Program (21YF1440300), the Shanghai Sailing Program (22YF1407700), and Key Disciplines of the Three-Year Action Plan for Strengthening the Public Health System in Shanghai (2023–2025; GWVI-11.1-14).

## AUTHOR AFFILIATIONS

[1]Department of Critical Care Medicine, Zhongshan Hospital, Fudan University, Shanghai, China

[2]Department of Critical Care Medicine, Shanghai Geriatric Medical Center, Shanghai, China

[3]Shanghai Institute of Infectious Disease and Biosecurity, Fudan University, Shanghai, China

[4]Nutrition Department, Zhongshan Hospital, Fudan University, Shanghai, China

## AUTHOR ORCIDs

Pan Jiang http://orcid.org/0009-0007-8981-5765
Jieqiong Song http://orcid.org/0000-0002-3445-8795
Ming Zhong http://orcid.org/0009-0005-5954-7195

## FUNDING

| Funder | Grant(s) | Author(s) |
|---|---|---|
| National Science Fund for Young Scholar | 82200061 | Shengyu Hao |
| Shanghai Sailing Program | 21YF1440300, 22YF1407700 | Shengyu Hao |
| Three-Year Action Plan for Strengthening the Public Health System in Shanghai | GWVI-11.1-14 | Ming Zhong |

## AUTHOR CONTRIBUTIONS

Shengyu Hao, Funding acquisition, Methodology, Writing – original draft | Chujun Zhou, Data curation, Funding acquisition, Methodology, Writing – original draft | Yilin Wei, Data curation, Investigation, Methodology, Writing – original draft | Yuxian Wang, Formal analysis, Investigation, Methodology, Writing – original draft | Pan Jiang, Formal analysis, Funding acquisition, Writing – review and editing | Jieqiong Song, Funding acquisition, Resources, Visualization, Writing – review and editing | Ming Zhong, Resources, Visualization, Writing – review and editing

## DATA AVAILABILITY

The data sets generated and analyzed during the current study are available in the NCBI Sequence Read Archive (SRA) repository, under the accession number PRJNA1328156.

## ETHICS APPROVAL

The study was approved by the Institutional Ethics Committee of Zhongshan Hospital, of Fudan University (approval code: B2023-016R). All procedures performed in this study involving human participants followed the Declaration of Helsinki (as revised in 2013). Written informed consent was obtained from each patient or their legally authorized representative (next of kin) if the patient lacked the capacity to provide consent due to their clinical condition.

## ADDITIONAL FILES

The following material is available online.

### Supplemental Material

**Supplemental figures (Spectrum00076-25-s0001.docx).** Fig. S1 and S2.
**Supplemental tables (Spectrum00076-25-s0002.docx).** Tables S1 and S2.

### Open Peer Review

**PEER REVIEW HISTORY (review-history.pdf).** An accounting of the reviewer comments and feedback.

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
