## [Reviewer comments · Microbiology Spectrum]

Microbiology Spectrum

Lower airway microbiota compositions and diversity among ventilator-associated pneumonia patients across COVID-19 epidemic phases: a retrospective study

Shengyu Hao, Chujun Zhou, Yilin Wei, Yuxian Wang, Pan Jiang, Jieqiong Song, and Ming Zhong

Corresponding Author(s): Ming Zhong, Zhongshan Hospital Fudan University

Review Timeline:

Submission Date:	January 9, 2025
Editorial Decision:	March 12, 2025
Revision Received:	April 21, 2025
Editorial Decision:	June 30, 2025
Revision Received:	August 12, 2025
Accepted:	August 14, 2025

Editor: Se-Ran Jun

Reviewer(s): Disclosure of reviewer identity is with reference to reviewer comments included in decision letter(s). The following individuals involved in review of your submission have agreed to reveal their identity: Daniel Fernández-Soto (Reviewer #2)

Transaction Report:

DOI: <https://doi.org/10.1128/spectrum.00076-25>

Re: Spectrum00076-25 (**Lower airway microbiota compositions and diversity among ventilator-associated pneumonia patients before, during, and after the COVID-19 pandemic: a retrospective study**)

Dear Dr. Ming Zhong:

Thank you for the privilege of reviewing your work. Below you will find my comments, instructions from the Spectrum editorial office, and the reviewer comments.

I received comments from one reviewer instead of two. Your revised manuscript will go through the revision process with the two reviewers if you submit a revision which addresses the comments provided.

Revision Guidelines

Sincerely,
Se-Ran Jun
Editor
Microbiology Spectrum

Reviewer #2 (Comments for the Author):

In this study, the authors analyze the microbiota composition of bronchoalveolar lavage fluid and serum samples from ventilator-associated pneumonia patients at intensive care units using next generation sequencing. They compare three groups of patients with different characteristics and from different time points during the pandemic, and find interesting differences.

Nevertheless, the study needs a proper definition of the hypothesis being tested. Most of the article, as stated in its title, suggests that it is evaluating the evolution of the microbiota throughout the pandemic. This would require that the three groups are somehow comparable but at different time points. On the contrary, the characteristics of the three groups as described in the methodology suggest that the results might be due to the evolution of the microbiota throughout an infection (from no infection, to acute and convalescent infection). Both aspects are related but are not the same, and they seem to be somehow confused along the article. It is of vital importance to clarify this in order to assess all authors' conclusions.

Here are some aspects that would help clarify the issue:

-A better definition of the study populations. The term pre-pandemic for samples up to January 2022 (almost two years after the declaration of the pandemic in March 2020) is questionable, and might be changed or otherwise well justified. Furthermore, the time window for the Dur and Post groups is not specified, but only described as "after the outbreak in Shanghai". Are they the same time window, but only differ in PCR positivity? Post-pandemic seem to suggest these samples were taken later than During-pandemic samples, is that so? If so, that should be specified in the methods or in Table 1 (and/or Supp Table 1).

-A better characterization of SARS-CoV-2 and COVID status of the three populations. Are all patients in the Pre group negative (PCR negative and/or seronegative?) for SARS-CoV-2? All patients in the Dur group seem to be admitted to ICU due to a severe COVID-19, but what is the case for the Post group? Were they admitted due to COVID-19 and are now PCR negative? Or they have a history of COVID-19 but were later admitted due to a different medical condition? And connected to this issue:

-Some more clinical information is needed. Table 1 and Supp Table 1 would benefit from including information about the cause for admission to ICU in the different groups, as this is relevant information when comparing them. It would also be necessary to include information as to whether the patients received antimicrobial treatment before sampling. If the objective of the study is to track the evolution of microbiota along time, the three populations should be similar for a proper comparison.

After addressing these points, once the nature of the comparison is properly defined (either tracking the progress of pandemic or the progress of an infection), the article should reflect that (title, abstract, discussion, conclusion...). Also the study limitations should probably be updated.

There are two additional points to be considered by the authors:

The methods section is very limited, and needs additional information. There does not seem to be any information on the processing of serum samples, as only BALF processing is referenced in the methods. Furthermore, no information is provided regarding the immune response characterization (how were those parameters measured?). Also, does "CD56" mean all CD56+ lymphocytes? Are they CD3+ or CD3-? Both?

The conclusion "our study highlights the differences in immune responses of VAP patients before, during, and after the COVID-19 pandemic" cannot be supported. The study highlights the differences in microbiota diversity, not so much the immune response (whose evaluation is very limited in this study).

Some additional minor points:

-Line 112 seems to suggest that centrifugation at 12,000 g for 5 min is enough to collect viral pathogens, which usually requires ultracentrifugation. Rephrasing is suggested.

-Lines 104-109 do not provide valuable information about the methodology, but rather general statements about the usefulness of NGS and as such could be deleted.

-Species in line 429 lack italica.

-It is suggested to include briefly the information of what each group (Pre/Dur/Post) means in the results section, and not only in the methods section, as to make the reading more fluid.

We appreciate the reviewer's thoughtful comments and suggestions, which have helped improve our study. Below, we address each question raised:

Reviewer 2

1. In this study, the authors analyze the microbiota composition of bronchoalveolar lavage fluid and serum samples from ventilator-associated pneumonia patients at intensive care units using next generation sequencing. They compare three groups of patients with different characteristics and from different time points during the pandemic, and find interesting differences.

Nevertheless, the study needs a proper definition of the hypothesis being tested. Most of the article, as stated in its title, suggests that it is evaluating the evolution of the microbiota throughout the pandemic. This would require that the three groups are somehow comparable but at different time points. On the contrary, the characteristics of the three groups as described in the methodology suggest that the results might be due to the evolution of the microbiota throughout an infection (from no infection, to acute and convalescent infection). Both aspects are related but are not the same, and they seem to be somehow confused along the article. It is of vital importance to clarify this in order to assess all authors' conclusions.

Responses: Thank you for your thoughtful feedback. We greatly appreciate your recognition of the study's potential contributions, particularly the application of next-generation sequencing to characterize microbiota dynamics in VAP patients across distinct epidemic phases. We fully agree that clarifying the study's hypothesis and differentiating between epidemic-phase effects and infection-status effects is essential. Below is our revised response incorporating your suggestions:

This article characterizes longitudinal shifts in the lower respiratory tract microbiota of VAP patients in ICU before, during, and after COVID-19 epidemic in Shanghai. While prior investigations have primarily focused on upper respiratory tract microbiota dynamics during acute SARS-CoV-2 infection, the impact of COVID-19 on lower airway microbial communities—especially during convalescence—remains underexplored. Emerging evidence suggests that SARS-CoV-2 infection disrupts

respiratory microbiota homeostasis, and accumulating data indicate persistent microbial alterations long after viral clearance. Ideal longitudinal designs would track microbiota changes within individual patients across infection phases (pre-infection → acute → recovery). Alternatively, contemporary cross-sectional comparisons could examine trios of never-infected, acutely infected, and recovered VAP patients matched for baseline characteristics. However, given the constraints of a retrospective single-center design and clinical realities, our study instead established three independent patient cohorts representing these infection states. Microbiota profiles were analyzed at the time of VAP diagnosis for: who were SARS-CoV-2-negative at the time of enrollment and had no prior COVID-19 diagnosis (Pre group), those who are currently infected with SARS-CoV-2 (Dur group), and those with resolved SARS-CoV-2 infection who tested negative during ICU stay (Post group) after developing VAP. In Table 1, we compared the baseline conditions of these patients and supplemented the reasons for their admission to the ICU. Except that the age was higher in the Dur group and the comorbidity index was lower in the Pre group, there were no statistical differences in other information between the Pre and Dur groups. Notably, while Pre and Dur groups demonstrated reasonable baseline comparability, the Post group exhibited demographic and clinical disparities that may confound microbiota interpretations. These limitations are explicitly addressed in the discussion section, acknowledging potential residual confounding by admission diagnosis and the need for future prospective studies with matched cohorts.

2. A better definition of the study populations. The term pre-pandemic for samples up to January 2022 (almost two years after the declaration of the pandemic in March 2020) is questionable, and might be changed or otherwise well justified. Furthermore, the time window for the Dur and Post groups is not specified, but only described as "after the outbreak in Shanghai". Are they the same time window, but only differ in PCR positivity? Post-pandemic seem to suggest these samples were taken later than During-pandemic samples, is that so? If so, that should be specified in the methods or in Table 1 (and/or Supp Table 1).

Responses: Thank you for your critical feedback on the study population definition. We agree that terminology and temporal alignment require clarification, and have made the following revisions:

We have replaced "pandemic" with "epidemic" throughout the manuscript to better reflect our single-center analysis in Shanghai, where local outbreak dynamics (the 2022 Shanghai COVID-19 wave) diverged from the global pandemic timeline due to government intervention. The study population has been redefined and the grouping criteria have been improved with a different time window for the three groups which they also differ in PCR positivity. They are three independent groups of people but not the evolution of the microbiota throughout the entire infection process. In the methodology section, we have refined the definition of population in order to assess the evolution of microbial communities throughout the entire pandemic period. A detailed description is provided in Revised_Manuscript_Clean_Version, from Line 93 to Line 105).

3. A better characterization of SARS-CoV-2 and COVID status of the three populations. Are all patients in the Pre group negative (PCR negative and/or seronegative?) for SARS-CoV-2? All patients in the Dur group seem to be admitted to ICU due to a severe COVID-19, but what is the case for the Post group? Were they admitted due to COVID-19 and are now PCR negative? Or they have a history of COVID-19 but were later admitted due to a different medical condition?

Response: Thank you for your valuable suggestion regarding to the characterization of SARS-CoV-2 and COVID status of the three populations. We have standardized and improved the characterization of SARS-CoV-2 and COVID-19 status for the three groups of enrolled patients. Patients were classified into three groups based on the timing of Shanghai's first COVID-19 lockdown, their medical history, and clinical examination results: Pre-epidemic group (Pre group): Patients admitted to the ICU from January 1, 2021 to February 28, 2022, before the first outbreak in Shanghai with negative real-time polymerase chain reaction (PCR) results and without any history of

SARS-CoV-2 infection; During-epidemic group (Dur group): Patients admitted after the outbreak in Shanghai from March 1, 2022 to December 31, 2022, with a confirmed COVID-19 diagnosis based on positive PCR testing of nasopharyngeal and/or oropharyngeal samples; Post-epidemic group (Post group): Patients admitted after the outbreak in Shanghai from January 1, 2023 to October 1, 2023, with a history of SARS-CoV-2 infection and negative PCR results upon current ICU admission. A detailed description is provided in the Revised_Manuscript_Clean_Version, from Line 93 to Line 105.

We have revised Table 1 and Supplementary Table 1 to explicitly document the primary indications for ICU admission. In the Pre group, all patients tested negative for SARS-CoV-2 upon hospital admission, as our institution implemented rigorous universal screening protocols during the pandemic. In the Dur group, not all patients were admitted to the ICU due to severe COVID-19. There were also patients with traumatic brain injury, those after cardiac surgery, patients with tumors, and those with connective tissue diseases, etc. Some of them were infected during their hospitalization, but they were confirmed to be PCR positive when admitted to the ICU. For the Post group, all patients had confirmed historical SARS-CoV-2 infection. Their current ICU admissions were unrelated to active COVID-19 infection, as confirmed by negative SARS-CoV-2 PCR results obtained at the time of ICU transfer and throughout their critical care course. To maintain infection control vigilance, any new pulmonary infiltrates, radiographic progression of pre-existing lung lesions, or febrile episodes during hospitalization prompted repeat COVID-19 testing and adherence to differential diagnostic algorithms, consistent with institutional protocols for managing emerging infectious diseases.

4. Some more clinical information is needed. Table 1 and Supp Table 1 would benefit from including information about the cause for admission to ICU in the different groups, as this is relevant information when comparing them. It would also be necessary to include information as to whether the patients received antimicrobial treatment before sampling. If the objective of the study is to track the evolution of microbiota along time, the three populations should be similar for

a proper comparison.

Response: We sincerely appreciate your constructive feedback. To address this concern, we have added ICU admission diagnosis for all patient groups in Table 1 and Supplementary Table 1. Furthermore, we have revised the Methods section. Based on previous studies, we used a sampling time window within 24 hours after the administration of antibiotics, and cited relevant references for support[1, 2]. We have provided a detailed description of these changes the Revised_Manuscript_Clean_Version, from Line 133 to Line 142, and the revised tracks are highlighted in the marked version submitted together with this response.

5. The methods section is very limited, and needs additional information. There does not seem to be any information on the processing of serum samples, as only BALF processing is referenced in the methods. Furthermore, no information is provided regarding the immune response characterization (how were those parameters measured?). Also, does "CD56" mean all CD56+ lymphocytes? Are they CD3+ or CD3-? Both?

Response: Thank you for your constructive feedback. We have addressed the concerns regarding the Methods section by expanding the description of serum sample processing and immune response characterization. For serum analysis, venous blood (approximately 2 mL) was collected in EDTA tubes and transported at room temperature. D-dimer was measured via automated coagulation analysis, while CRP, PCT, and ferritin were quantified using immunoturbidimetric assays. Serum cytokines (TNF- α , IL-1 β , IL-2R, IL-6, IL-8, IL-10) were analyzed by ELISA with R&D Systems kits according to the manufacturer's protocols.

Regarding immune profiling, whole blood samples were processed within 48 hours using flow cytometry. Leukocyte differentials were performed to ensure lymphocyte concentrations were within the instrument's linear range. Lymphocyte subsets were defined as follows: T cells (CD3⁺), B cells (CD3⁻CD19⁺), helper T cells (CD3⁺CD4⁺), cytotoxic T cells (CD3⁺CD8⁺), and NK cells (CD3⁻CD16⁺CD56⁺). These were

analyzed using a FACSCalibur cytometer with BD Multitest IMK reagents, including two antibody cocktails: CD3-FITC/CD8-PE/CD45-PerCP/CD4-APC and CD3-FITC/CD16⁺CD56⁺-PE/CD45-PerCP/CD19-APC, alongside with erythrocyte lysing solution. A detailed description is provided in the Revised_Manuscript_Clean_Version, from Line 107 to Line 128.

To clarify CD56 interpretation, NK cells were specifically gated as CD3⁻CD16⁺CD56⁺. All methods are now detailed in the revised manuscript, with tracked changes highlighted. In addition, we have made supplements to the part related to cell subsets in Table 1 and supplementary Table 1. Thank you for your suggestions, which have strengthened the study's rigor.

6. The conclusion "our study highlights the differences in immune responses of VAP patients before, during, and after the COVID-19 pandemic" cannot be supported. The study highlights the differences in microbiota diversity, not so much the immune response (whose evaluation is very limited in this study).

Response: Thank you for your critical insight. After careful reassessment, we have deleted the sentence in question to align the conclusion with the study's actual focus on microbiota diversity.

7. Some additional minor points.

- 7.1 Line 112 seems to suggest that centrifugation at 12,000 g for 5 min is enough to collect viral pathogens, which usually requires ultracentrifugation. Rephrasing is suggested.

Response: We sincerely appreciate your meticulous review and the valuable comment. Your feedback is instrumental in improving the quality of our manuscript.

Upon re-evaluating the sample collection and metagenomic workflow, we've made revisions to the method section. You are correct that viral pathogens typically require specialized techniques like ultracentrifugation for collection. However, since viral pathogens are predominantly present within cells and there is a relatively small amount in the supernatant, centrifugation at 12,000 g is sufficient for our purposes.

This centrifugation force effectively pellets the components of interest for subsequent DNA extraction. We also noticed an error in the centrifugation time stated previously. It should be 10 minutes instead of 5 minutes, and we have corrected this in the manuscript. To further strengthen the validity of our methods, we have incorporated relevant references that support our experimental approach[3, 4].

Moreover, we have supplemented the description of blood sample collection and processing. The updated content can be found in the method section, specifically in Line 129 to Line 145 of the revised manuscript.

7. 2 Lines 104-109 do not provide valuable information about the methodology, but rather general statements about the usefulness of NGS and as such could be deleted.

Response: Thank you for your feedback. We sincerely appreciate your suggestion to streamline the Methods section. In response, we have deleted Lines 104-109, which contained general statements about NGS utility.

7. 3 Species in line 429 lack italica.

Response: Thank you for pointing out this formatting oversight. We have made the correction in Line 391 of the revised manuscript by italicizing the species name.

7. 4 It is suggested to include briefly the information of what each group (Pre/Dur/Post) means in the results section, and not only in the methods section, as to make the reading more fluid.

Response: Thank you for your constructive suggestion regarding manuscript organization. We have added the group definitions at the beginning of the Results section (Line 222 to Line 229 of the revised manuscript) as follows:

"Patients were stratified into three groups based on admission timing and SARS-CoV-2 infection status. The Pre-group (n=58, 33.1%) included individuals admitted to the ICU prior to Shanghai's first confirmed SARS-CoV-2 outbreak, with negative SARS-CoV-2 PCR results and no documented history of infection. The Dur-group (n=58, 33.1%) consisted of patients admitted during the local outbreak

period, who tested positive for SARS-CoV-2 via nasopharyngeal/oropharyngeal PCR. The Post-group (n=59, 33.8%) comprised patients admitted following the outbreak's resolution, with a history of SARS-CoV-2 infection but negative PCR results upon current ICU admission."

We appreciate your insightful comment that significantly improves the manuscript's clarity.

Reference

1. Vaughn, V.M., et al., *Excess Antibiotic Treatment Duration and Adverse Events in Patients Hospitalized With Pneumonia: A Multihospital Cohort Study*. *Ann Intern Med*, 2019. **171**(3): p. 153-163.
2. Sun, T., et al., *A Paired Comparison of Plasma and Bronchoalveolar Lavage Fluid for Metagenomic Next-Generation Sequencing in Critically Ill Patients with Suspected Severe Pneumonia*. *Infect Drug Resist*, 2022. **15**: p. 4369-4379.
3. Wang, L., et al., *Metatranscriptome of human lung microbial communities in a cohort of mechanically ventilated COVID-19 Omicron patients*. *Signal Transduct Target Ther*, 2023. **8**(1): p. 432.
4. Xu, F., et al., *Impact of metagenomics next-generation sequencing on etiological diagnosis and early outcomes in sepsis*. *J Transl Med*, 2025. **23**(1): p. 394.

Re: Spectrum00076-25R1 (**Lower airway microbiota compositions and diversity among ventilator-associated pneumonia patients across COVID-19 epidemic phases: a retrospective study**)

Dear Dr. Ming Zhong:

Thank you for the privilege of reviewing your work. Below you will find my comments, instructions from the Spectrum editorial office, and the reviewer comments.

Revision Guidelines

Sincerely,
Se-Ran Jun
Editor
Microbiology Spectrum

Reviewer #2 (Comments for the Author):

The authors have thoroughly addressed all previous concerns. The clarity and the overall quality of the paper have significantly improved, specially with the improved methods section and the detailed supplementary table.

Just one final minor comment:

-The authors seem to delete Table 1 and include all the information in Supplementary Table 1. The decision is endorsed, but there are still some references to the now deleted Table 1 in the text that should be removed.

Reviewer #3 (Comments for the Author):

This manuscript by Hao et al. presents a timely and clinically relevant investigation into how the composition and diversity of the lower airway microbiota in VAP patients varied across different phases of the COVID-19 epidemic in Shanghai. Using NGS of BALF and serum samples from 175 ICU patients, the authors aimed to characterize microbial shifts and their potential association with patient outcomes, including ICU mortality.

To strengthen the manuscript and enhance its scientific rigor and reproducibility, the following clarifications and expansions are recommended:

- The authors reference "established criteria" for diagnosing VAP but do not specify which criteria were used. The manuscript should clearly define the diagnostic criteria (e.g., clinical signs, imaging findings, microbiological results) and describe how these were operationalized in this study.
- While it is noted that samples were collected within 24 hours of antibiotic administration, further detail is needed. Please clarify whether patients were on empiric or targeted therapy, the classes of antibiotics administered, and how antibiotic exposure was controlled for or incorporated into the microbiome analyses.
- The manuscript does not mention the use of negative controls (e.g., blank extraction controls) or mock communities to assess for reagent or environmental contamination in the NGS workflow. This is critical for accurate interpretation of results. Please describe the contamination control measures used.
- Information on microbial α -diversity metrics is currently found only in figure legends. These details-including which indices were used (e.g., Shannon, Simpson), and which software or analytical pipelines were applied-should be clearly described in the methods section.
- Given the multi-year duration of the study and the potential for sequencing batch effects, it is important to state whether batch effects were evaluated or corrected for in the microbiome analysis.
- The flow cytometry methods lack essential details. While the antibody panels are provided, the manuscript does not include information about gating strategies, quality control measures (e.g., use of isotype or FMO controls), or whether results are reported as absolute counts or percentages. Including a representative gating strategy figure would improve clarity and reproducibility.
- The reported detection of HSV-1 and *Aspergillus fumigatus* raises the question of validation. Were these microbial loads confirmed by orthogonal methods such as qPCR, or are they based solely on computational NGS data? Clarification on how these findings were validated would strengthen the reliability of the conclusions.
- The methods state that informed consent was obtained from "parents or legal guardians," which appears inconsistent with an adult study population (age {greater than or equal to}18). Please revise or clarify this point.
- the discussion section would benefit from a more balanced interpretation of the findings. The manuscript frequently attributes microbial shifts to SARS-CoV-2 infection, but does not adequately address other potential confounding factors, such as changes in ICU admission criteria, patient case mix, antimicrobial stewardship practices, and nosocomial transmission risk, that may also influence microbial composition. These should be discussed in greater depth.

We appreciate the reviewer's thoughtful comments and suggestions, which have helped improve our study. For clarity, all line numbers cited in our responses refer to the file named 'Revised_Manuscript_Clean_Version'. Below, we address each question raised:

Reviewer 2

The authors have thoroughly addressed all previous concerns. The clarity and the overall quality of the paper have significantly improved, specially with the improved methods section and the detailed supplementary table.

Just one final minor comment:

-The authors seem to delete Table 1 and include all the information in Supplementary Table 1. The decision is endorsed, but there are still some references to the now deleted Table 1 in the text that should be removed.

Responses: Thank you for your careful review of our revised manuscript and for your positive feedback regarding the improvements made. We sincerely appreciate your time and valuable insights. We have now carefully removed remaining references to the deleted Table 1 throughout the manuscript text. The relevant information is correctly located and referenced in Supplementary Table 1

Reviewer 3

1. The authors reference "established criteria" for diagnosing VAP but do not specify which criteria were used. The manuscript should clearly define the diagnostic criteria (e.g., clinical signs, imaging findings, microbiological results) and describe how these were operationalized in this study.

Responses: Thank you for your critical feedback on the study VAP definition. The definition and diagnosis of VAP remain controversial, with substantial variations in reference standards. Histopathological examination of inflamed and infected lung tissue is considered the most accurate diagnostic tool for diagnosing VAP. However, a lung biopsy is not practical or appropriate for the critically ill, and post-mortem examination is too late to influence treatment[1, 2]. Clinical guidelines, therefore,

attempt to amalgamate clinical, radiological and microbiological criteria to support the diagnosis of VAP. Clinical examination is essential but has a sensitivity of 66.4% (95% CI 40.7 – 85.0) and specificity of 53.9% (95% CI 34.5 – 72.2) in the diagnosis of VAP[3]. We have added the statements from Line 87 to Line 101 in Method part:

Briefly, the clinical diagnostic criteria are as follows: (1) fever $> 38^{\circ}\text{C}$ with no other cause, and (2) leukocytosis or leukopenia and at least one of the following: (1) new onset or change in sputum, or (2) cough, dyspnea or tachypnoea, or (3) worsening gas exchange. The diagnostic criteria for radiology are as follows: chest radiographs or computed tomograms with evidence of pulmonary infiltrates OR air bronchograms. If there is underlying pulmonary disease, it is necessary to compare with previous imaging to confirm new or progressive changes. The diagnostic criteria for microbiology are as follows: positive quantitative culture from minimally contaminated lower respiratory tract specimen or positive sputum culture or non-quantitative lower respiratory tract culture[4]. In our study, for patients who underwent endotracheal intubation in the ICU for more than 48 hours, if both clinical and radiology diagnostic criteria were met, we highly suspected VAP. Sputum culture and BALF were collected for testing at the same time. If either sputum culture or BALF were positive, the diagnosis of VAP was confirmed.

2. While it is noted that samples were collected within 24 hours of antibiotic administration, further detail is needed. Please clarify whether patients were on empiric or targeted therapy, the classes of antibiotics administered, and how antibiotic exposure was controlled for or incorporated into the microbiome analyses.

Responses: Thank you for your valuable feedback and the opportunity to clarify our methodology regarding antibiotic administration.

As we state in the manuscript, our protocol was designed to minimize the immediate impact of treatment on the results: "To mitigate antibiotic effects on microbial profiling, our protocol adheres to the recommended practice of specimen collection

within 24 hours of antibiotic administration."

To control for antibiotic exposure as a potential confounding variable, we performed a statistical analysis comparing the classes of antibiotics administered across the Pre, During, and Post groups (Supplementary Table 2). Our analysis confirmed that the antibiotic regimens were largely comparable across the groups. The most frequently administered antibiotic classes were carbapenems, third-generation cephalosporins, and glycopeptides. Statistical testing showed no significant differences in the distribution of most individual drug classes among the three groups. While the overall comparison for Combination therapy showed a statistically significant difference ($p=0.048$), subsequent pairwise comparisons between the groups revealed no significant differences. We have detailed these revisions in the method section (Line 243-245) and results section (Line 305-314) of the clean manuscript. All changes are highlighted in the track-changed version submitted with this response.

In summary, by collecting samples shortly after the initial dose and confirming the statistical similarity of antibiotic regimens across the study groups, we have controlled for the potential confounding effects of antibiotic exposure in our microbiome analyses.

3. The manuscript does not mention the use of negative controls (e.g., blank extraction controls) or mock communities to assess for reagent or environmental contamination in the NGS workflow. This is critical for accurate interpretation of results. Please describe the contamination control measures used.

Response: We highly appreciate this suggestion and have supplemented the relevant content in the Methods section (Lines 194-198) as follows: For each batch of NGS workflows, blank samples are included and subjected to the identical procedures as the sequencing samples (from nucleic acid extraction to library preparation and sequencing). During data analysis, the NGS detection background is compared with this negative control to minimize the impact of environmental contamination.

4. Information on microbial α -diversity metrics is currently found only in figure legends. These details-including which indices were used (e.g., Shannon,

Simpson), and which software or analytical pipelines were applied-should be clearly described in the methods section.

Response: Thank you for your valuable feedback. We agree that the details regarding the α -diversity analysis should be clearly described in the Methods section. We have now added this information to the 'Statistical analysis' subsection of the Methods. This information can be found on Lines 237-238 of the revised manuscript. We have now incorporated a description of the Linear Discriminant Analysis Effect Size (LEfSe) method into the "Statistical analysis" subsection of the Methods. This information can be found on Lines 238-242 of the revised manuscript.

5. Given the multi-year duration of the study and the potential for sequencing batch effects, it is important to state whether batch effects were evaluated or corrected for in the microbiome analysis.

Response: Thank you for raising this important point. We recognize that potential batch effects are a critical consideration in a study spanning multiple years. We employed a multi-faceted strategy to minimize and control for such effects, which we have now detailed in the Methods section (Line 221-229).

Our approach included:

Standardized Processing: All samples were processed and sequenced at a single certified facility (Center for Infectious Diseases, Vision Medicals Co., Ltd, Guangzhou, China) using standardized protocols to ensure high procedural consistency and minimize inter-batch variability. **Reagent and Environmental QC:** Each new batch of sequencing reagents was pre-emptively assessed to identify and characterize any lot-specific microbial signatures. Negative controls (sequencing blanks) were included in every sequencing run to monitor for environmental and reagent-derived contaminants. This background profile was then bioinformatically subtracted from all patient samples. **Internal Standardization:** As described in the manuscript, a spiked internal standard was added to each sample prior to DNA extraction. This allowed for the monitoring of technical consistency and extraction

efficiency across different batches.

Collectively, these stringent quality control measures at the procedural, reagent, and bioinformatic levels were implemented to ensure that the observed differences in the microbiome were biological in origin and not due to technical artifacts from batch-to-batch variation.

6. The flow cytometry methods lack essential details. While the antibody panels are provided, the manuscript does not include information about gating strategies, quality control measures (e.g., use of isotype or FMO controls), or whether results are reported as absolute counts or percentages. Including a representative gating strategy figure would improve clarity and reproducibility.

Response: Thank you for your insightful comments. We deeply appreciate your meticulous attention to the nuances of the flow cytometry methods. The flow cytometry experiments in this study were conducted at the Clinical Laboratory of our hospital. Initially, we consulted the methodologies and reagents for cell subset detection as delineated in previous publications from our hospital's clinical laboratory. Nevertheless, through further communication with the laboratory physician in charge of the flow cytometry testing, we ascertained that both the detection reagents and gating strategies had been updated specifically for the samples incorporated in this study.

To address the concerns raised, we have revised and supplemented the relevant content in the Methods section (Line 137 to Line 151) as follows: “The BD Multitest™ 6-Color TBNK (Catalog No. 662967) included antibody cocktails as follows: BD Multitest CD3 FITC/CD8 APC-Cy7/CD45 PerCP-Cy5.5/CD4 PE-Cy7/CD16⁺CD56⁺ PE/CD19 APC. For whole - blood samples from patients collected in EDTA blood collection tubes, the procedure was carried out per the product instructions. Briefly, 20 μL of the diluted BD Multitest™ 6 - Color TBNK reagent was pipetted to the bottom of the tube. Then, 50 μL of well - mixed anticoagulated whole blood was pipetted to the tube bottom, followed by incubation at room temperature in the dark

for 30 minutes. Next, 450 μ L of 1 \times BD FACS™ Lysing Solution (Catalog No. 349202) was added to the tube. After shaking well, the tube was incubated in the dark at room temperature for 30 minutes. The samples were then ready for analysis on the flow cytometer. Unstained samples from the same blood specimen of the same patient were used as negative controls. The gating strategy and representative flow cytometry plots are provided in Supplementary Figure 1. The statistical analysis presents the percentage of each cell subset within the CD45⁺ lymphocyte population, and the results are shown in Supplementary Table 1.”

The gating strategy used in the assay was provided by the Clinical Laboratory. Representative images are presented in Supplementary Figure 1, with the specific gating strategy detailed in the legend of Supplementary Figure 1 as follows:

Supplementary Fig 1. Gating strategy for lymphocyte subpopulations. A. Identification of CD45⁺ lymphocytes from total events. B. CD3⁺ T cells and CD3⁻ non-T cells gated within CD45⁺ lymphocytes. C. Subsetting of CD3⁺ T cells into CD4⁻CD8⁻ T helper (Th) cells and CD4⁻CD8⁺ cytotoxic T (Tc) cells using CD4 vs. CD8. D. Gating of CD19⁺ B cells from CD3⁻ non-T cells. E. Discrimination of B cells (CD19⁺) and NK cells (CD19⁻CD16⁺CD56⁺) via CD19 vs. CD16/CD56.

Unstained samples from the same blood specimen of the same patient were used as negative controls. Results are expressed as of each cell subset within the CD45⁺ lymphocyte population, with the unit (%) indicated in Supplementary Table 1.

7. The reported detection of HSV-1 and *Aspergillus fumigatus* raises the question of validation. Were these microbial loads confirmed by orthogonal methods such as qPCR, or are they based solely on computational NGS data? Clarification on how these findings were validated would strengthen the reliability of the conclusions.

Response: Thank you for your insightful question. Regarding the detection of HSV-1 and *Aspergillus fumigatus*, we would like to clarify that in the present study, their microbial loads were determined based on computational analysis of next-generation sequencing (NGS) data. We have supplemented the Methods section (Lines 193-194) with the following content: "Microbial loads in this research were determined based

on bioinformatic analysis of NGS data."

Notably, numerous studies have confirmed that metagenomic next-generation sequencing (mNGS) exhibits comparable performance to qPCR in terms of sensitivity and specificity. To address this point, we have added the following statement in the Discussion section (Lines 434-436): "with studies demonstrating that its diagnostic performance for respiratory pathogens is comparable to that of targeted qPCR assays[5-8]".

8. The methods state that informed consent was obtained from "parents or legal guardians," which appears inconsistent with an adult study population. Please revise or clarify this point.

Response: Thank you for pointing out this important inconsistency. You are correct that our study population is adults, and the original phrasing was not precise. We have revised the sentence in the Methods section for clarity.

The revision now reads: " Written informed consent was obtained from each patient or their legally authorized representative (next-of-kin) if the patient lacked the capacity to provide consent due to their clinical condition."

As is common in research involving critically ill ICU patients, many participants were sedated, intubated, or otherwise unable to provide consent themselves. In such cases, and in accordance with our institutional ethical guidelines, consent was obtained from a legally authorized representative or next-of-kin. We believe the revised wording accurately reflects our procedure. The change has been made in Method section (Line 257-259).

9. The discussion section would benefit from a more balanced interpretation of the findings. The manuscript frequently attributes microbial shifts to SARS-CoV-2 infection, but does not adequately address other potential confounding factors, such as changes in ICU admission criteria, patient case mix, antimicrobial stewardship practices, and nosocomial transmission risk, that may also influence microbial composition. These should be discussed in greater depth.

Response: Thank you for this valuable suggestion. We fully agree that a more balanced interpretation of the findings is essential, and we acknowledge that the initial discussion did not sufficiently address potential confounding factors that might influence microbial composition. We have added a supplementary statement in the Discussion section (Lines 542-559) as follows:

While our findings demonstrate significant shifts in the lower airway microbiota across the pandemic phases, these results must be interpreted within the context of multiple, co-evolving clinical and epidemiological factors. Although ICU admission criteria could have theoretically shifted, the overall baseline severity of illness was comparable across the cohorts, as evidenced by non-significant differences in APACHE II and SOFA scores. However, critical demographic and clinical differences likely acted as confounders. Specifically, patients in the During-epidemic group were older and had a higher comorbidity burden as measured by the Charlson score, while the Post group had significantly higher rates of steroid or immunosuppressive drug use, all of which can independently shape the airway microbiome. Regarding antimicrobial pressure, our analysis revealed that the use of most individual antibiotic classes was largely comparable across the groups, suggesting this may not have been the primary driver of the observed microbial changes. Furthermore, pandemic-related surges may have impacted nosocomial transmission dynamics for common VAP pathogens [9]. Therefore, while SARS-CoV-2 infection appears to be a key factor, particularly in reducing microbial diversity, the observed shifts in the VAP microbiome must be understood as a multifactorial phenomenon reflecting a complex interplay between the virus, evolving patient characteristics, clinical practices, and the ICU environment.

References

1. Al-Omari, B., et al., *Systematic review of studies investigating ventilator associated pneumonia diagnostics in intensive care*. BMC Pulm Med, 2021. **21**(1): p. 196.
2. Howroyd, F., et al., *Ventilator-associated pneumonia: pathobiological heterogeneity*

- and diagnostic challenges. Nat Commun, 2024. 15(1): p. 6447.*
3. Fernando, S.M., et al., *Diagnosis of ventilator-associated pneumonia in critically ill adult patients-a systematic review and meta-analysis. Intensive Care Med, 2020. 46(6): p. 1170-1179.*
 4. Fenn, D., et al., *Composition and diversity analysis of the lung microbiome in patients with suspected ventilator-associated pneumonia. Crit Care, 2022. 26(1): p. 203.*
 5. Goelz, H., et al., *Next- and Third-Generation Sequencing Outperforms Culture-Based Methods in the Diagnosis of Ascitic Fluid Bacterial Infections of ICU Patients. Cells, 2021. 10(11).*
 6. Jiang, Z., et al., *Clinical performance of metagenomic next-generation sequencing for diagnosis of pulmonary Aspergillus infection and colonization. Front Cell Infect Microbiol, 2024. 14: p. 1345706.*
 7. Zhang, P., et al., *Clinical application of targeted next-generation sequencing in severe pneumonia: a retrospective review. Crit Care, 2024. 28(1): p. 225.*
 8. He, Y., et al., *Enhanced DNA and RNA pathogen detection via metagenomic sequencing in patients with pneumonia. J Transl Med, 2022. 20(1): p. 195.*
 9. Cooper, B.S., et al., *The burden and dynamics of hospital-acquired SARS-CoV-2 in England. Nature, 2023. 623(7985): p. 132-138.*

Re: Spectrum00076-25R2 (**Lower airway microbiota compositions and diversity among ventilator-associated pneumonia patients across COVID-19 epidemic phases: a retrospective study**)

Dear Dr. Ming Zhong:

Your manuscript has been accepted, and I am forwarding it to the ASM production staff for publication. Your paper will first be checked to make sure all elements meet the technical requirements. ASM staff will contact you if anything needs to be revised before copyediting and production can begin. Otherwise, you will be notified when your proofs are ready to be viewed.

Sincerely,
Se-Ran Jun
Editor
Microbiology Spectrum